# Efficacy of Penicillin–Streptomycin Brands against *Staphylococcus aureus*: Concordance between Veterinary Clinicians’ Perception and the Realities

**DOI:** 10.3390/antibiotics12030570

**Published:** 2023-03-14

**Authors:** Takele Beyene Tufa, Asegid Guta, Tafese B. Tufa, Dereje Nigussie, Ashenafi Feyisa Beyi, Fanta D. Gutema, Fikru Regassa

**Affiliations:** 1College of Veterinary Medicine and Agriculture, Addis Ababa University, Bishoftu P.O. Box 34, Ethiopia; 2College of Health Sciences, Arsi University, Asella P.O. Box 193, Ethiopia; 3Hirsch Institute of Tropical Medicine, Arsi University, Asella P.O. Box 193, Ethiopia; 4Ethiopian Public Health Institute, Addis Ababa P.O. Box 1242, Ethiopia; 5College of Veterinary Medicine, Iowa State University, Ames, IA 50011, USA; 6Department of Occupational and Environmental Health, University of Iowa, Iowa City, IA 50011, USA

**Keywords:** antibiotic brand, penicillin–streptomycin, efficacy, *S. aureus*, antimicrobial susceptibility, perception, Ethiopia

## Abstract

Antibiotics must be safe and effective for use in both human and veterinary medicine. However, information about the efficacy of different brands of antibiotics commonly used in veterinary practices is lacking in Ethiopia. In this study, we determined the efficacy of three brands of penicillin–streptomycin (Pen&strep, Penstrep, and Pro&strep) by performing antimicrobial susceptibility testing against *Staphylococcus aureus* isolated from cow milk from dairy farms in the towns of Sebata and Bishoftu, Central Ethiopia. We also assessed the knowledge, attitudes, and practices (KAP) of veterinarians regarding the quality and use of brand antibiotics and the antibiotic utilization practices of dairy farm personnel using a structured questionnaire. Of 43 *S. aureus* isolated and tested, 33 (77%), 10 (23%), and 1 (2%) were susceptible to brands A, B, and C, respectively. According to the respondents, all of them reported that penstrep is the most prescribed antibiotic in dairy farms (100%), followed by oxytetracycline (78%) and sulfa drugs (52%). All veterinarians perceived that antibiotics imported from Western countries have a higher efficacy than those from Eastern countries, and they preferred brand A to the other brands, witnessing its better clinical outcome. The majority (87%) and a little more than half (53%) of the respondents perceived the overuse of antibiotics in veterinary clinics and dairy farms, respectively. Our study revealed the better efficacy of brand A against *S. aureus* compared to the other brands. Interestingly, the veterinarians’ perception of and preference toward the use of brand antibiotics agreed with the findings of our antibacterial susceptibility testing. The prudent use of brand A is critically important for sustaining effective treatment, avoiding the risk of antimicrobial resistance, and helping to address animal welfare issues.

## 1. Introduction

Antibiotics play a significant role in reducing morbidity and mortality associated with infectious diseases in animals and humans [1]. In food-producing animals, antibiotics are used to control, prevent, and treat infections and enhance animal growth and productivity by improving feed conversion efficiency [2,3]. β-lactams, tetracyclines, aminoglycosides, lincosamides, macrolides, and sulfonamides are the most commonly used antimicrobials in food-producing animals [4]. Penicillin, a β-lactam antibiotic, is predominantly used for its efficacy against Gram-positive bacteria [5], while streptomycin-sulfate, an aminoglycoside antibiotic, is used for its effect against Gram-negative bacteria and *Mycobacteria* spp. [6]. “Penstrep” is an antibacterial suspension comprised of a fixed penicillin and streptomycin combination. It is widely used in the control, prevention, and treatment of infections in animals, especially in food-producing domestic animals because of its combined efficacy against both Gram-positive and Gram-negative bacteria for empirical treatment [6].

Different antibiotics could potentially have variations in their efficacy against the microorganisms they interact with. Differences in efficacy could occur between different groups of antibiotics and within the same antibiotic with different brands from different manufacturers, ascribed to the variation in the quality and quantity of active ingredients. Low-quality medicines could also be attributed to the packaging, transportation, and storage conditions as well as distribution systems [7,8]. Administering counterfeit drugs could result in therapeutic failure, toxicity, allergic reactions, the development of antimicrobial resistance (AMR), prolonged illness, a high cost of treatment, and even mortality [8,9].

The quality of drugs used in most developing countries is questionable, although evidence is largely anecdotal. A study published in *The Lancet* indicated that fake and poor-quality drugs are alarmingly common in some African and Southeast Asian countries [10,11]. Other studies substantiated that fake and low-quality medicines are prevalent in the developing world; substandard and falsified medicines rose to almost 19% in African countries in 2018 [12,13]. The World Health Organization (WHO) data indicated that about 10% of medications in the poorest countries are fraudulent or substandard [13,14]. A recent report by the United Nations Office of Drug Crime (UNODC) has indicated that fake medicines kill almost 500,000 Sub-Saharan Africans per year [15]. The reasons for poor quality include the widespread counterfeiting of medicines, the decomposition of the active ingredients due to improper storage, and poor quality assurance during the manufacture of drugs [8,16,17,18]. Most developing and underdeveloped countries suffer from the direct and indirect effects of poor-quality drugs to a high degree [17].

In Ethiopia, most livestock infections are treated empirically based on clinical signs and clinicians’ experience, which potentially creates antimicrobial selection pressure and leads to the emergence of resistant bacterial strains [19]. Several studies reported low to high levels of AMR against the majority of essential antimicrobial agents currently approved for use in human and veterinary medicine [20,21,22,23,24,25,26]. Our prior studies demonstrated that the prescription and utilization of antibiotics both in veterinary clinics and animal farms are often irrational in Ethiopia [19,26,27]. This, combined with the currently available different brands of antimicrobial agents with unknown efficacy, makes the selection of an appropriate agent a challenging task. This highlights the importance of in vitro antimicrobial susceptibility testing (AST) in providing evidence about drug efficacy, improving veterinary services, and curbing the problem of AMR [28,29].

Within Ethiopia, information on the efficacy and quality of various brands of veterinary antibiotics in general and penstrep in particular is lacking. However, this is critically needed for a clinical decision to select the most effective drugs. Hence, in this study, we determined the efficacy of different brands of penstrep against *S. aureus* isolated from cow milk and assessed the knowledge, attitudes, and practices (KAP) of veterinarians regarding the quality and use of brand antibiotics and antibiotic utilization practices of farm personnel in dairy farms in Sebata and Bishoftu, Central Ethiopia. The findings from the study on the efficacy of leading brands of penstrep used in Ethiopia could give insights into the quality of products, thereby helping clinicians to make an evidence-based decision in selecting an appropriate brand for the effective treatment of infections in animals. This will also help to tackle animal welfare concerns and minimize the current global burden of AMR in the livestock sector.

## 2. Results

### 2.1. Bacterial Isolation and Antimicrobial Susceptibility Testing

Of the 209 milk samples examined, 43 (21%) samples yielded *S. aureus*. The mean zone of inhibition (ZI) recorded for all 43 *S. aureus* tested against the three brands was 16.86 mm (SD ± 3.16), 9.92 mm (SD ± 2.10), and 6.20 mm (SD ± 2.24) for brands A, B, and C, respectively (Figure 1). An extremely low mean ZI was recorded for standard discs of penicillin G (2.193, SD ± 2.21) compared with the streptomycin (10.84, SD ± 3.56). A Kruskal–Wallis rank sum test showed that there was a statistically significant difference in the ZI of *S. aureus* growth between different brands of penstrep and standard antibiotics (S and PEN) (Chi-squared = 158.1, df = 4, *p*-value ≤ 0.001).

Compared to the standard set for the penicillin disc (10 µg, Oxoid), all of the 43 *S. aureus* tested were resistant to all brands. However, 81% (35/43) of the isolates were susceptible to brand A, and all of them were resistant to both brand B and brand C compared to the standards set for the streptomycin disc (10 µg, Oxoid). The study also indicated that out of the 43 *S. aureus* tested against the standard discs, 98% (*n* = 42) of isolates were resistant to penicillin, whereas 63% (*n* = 27) and 21% (*n* = 9) of isolates showed resistance and IR to streptomycin, respectively (Table 1).

### 2.2. Farm Antibiotic Utilization Practices

According to the respondents, the usage of penstrep among dairy farms is ranked first (100%), followed by oxytetracycline (78%, 18/23) and sulfa drugs (52%, 12/23). In all the dairy farms (100%, 23), penstrep was used at least once within the last 3 months. Penstrep was the most commonly prescribed antibiotic on the study dairy farms for treatment of septicemic conditions (100%), wounds (100%), all suspected systemic bacterial infections (74%, 17/23), and mastitis cases that did not respond to intramammary antibiotic infusions (39%, 9/23). It was also used for both prophylaxis and metaphylaxis during viral infections such as foot and mouth disease (FMD) (57% and 39%) and lumpy skin disease (LSD) outbreaks (57% and 35%), respectively. Oxytetracycline was the second most extensively prescribed antibiotic in the dairy farms in the study area (87%, 20/23) for treating unknown diseases in non-pregnant and non-lactating cows. However, the long-acting oxytetracycline formulation (20%) was the most prescribed antibiotic for treating secondary bacterial complications in dairy animals infected with viral diseases, namely, FMD and LSD, and for prophylaxis (78%, 18/23) and metaphylaxis use (91%, 21/23). Sulfa drugs, the third most widely prescribed antibiotics in the dairy farms in the study area, were commonly prescribed for managing diarrhea in calves (52%, 12/23).

### 2.3. Knowledge, Perception, and Practices of Veterinarians

All veterinarians (100%) agreed and perceived that antibiotics imported from Western countries have better quality than those imported from Eastern countries. Similarly, all of them perceived that some antibiotics of poor quality are available on the local market. Although some veterinarians agreed that they prescribed antibiotics by international nonproprietary name (47%, 14/30) and by brand name (67%, 20/30), they also agreed that generic drugs are perceived as equivalent to brand drugs (70%, 21/30) compared to generic antibiotics perceived as substandard drugs (30%, 9/30) (Table 2).

Veterinarians’ perception towards the country of antibiotics manufacturers revealed that all the participants agreed that penstrep produced in the UK is of better quality than that produced in China. Similarly, all the participants perceived that there are differences in clinical outcomes among the three different brands of penstrep. Though brand A penstrep is relatively more expensive, all (30, 100%) of the veterinarians preferred using it, noting that brand A has better clinical effectiveness than the other two brands (Table 2).

Veterinarians’ knowledge about antibiotic use indicated that all veterinarians have good knowledge about the major clinical indications of antibiotics (Table 2). However, 87% (26/30) and 53% (16/30) of them perceived that antibiotics are overused in veterinary clinics and dairy farms, respectively.

## 3. Discussion

To the best of our knowledge, this study is the first report that evaluated the efficacy of different brands of antibiotics used in veterinary medicine and assessed the KAP of veterinarians regarding the quality and uses of different brands of antibiotics and farm personnel regarding antibiotic utilization practices in Ethiopia. Our study demonstrated variability in the efficacy of the three brands of penstrep, which is widely used for the treatment of infections in animals.

Our study showed a considerable difference in efficacy among the three brands, where brand A had better efficacy over the other brands compared with the standard streptomycin. The variation could be because of the difference in the quality of active ingredients present in the formulation or logistic systems. This could be due to the difference in their manufacturing processes and/or the amount of active ingredients in the drugs [8]. Another possible reason could be the high concentration of streptomycin in brand A (250 mg/mL) compared to other brands (200 mg/mL), making it the desired dose level for effectively killing *S. aureus* strains. The difference in the quality could result in a significant difference in the price of drugs, by which the prices of drugs with higher efficacy are expensive compared to the lower-efficacy drugs [30]. This implies the selection of poor-quality drugs at low prices by poor farmers, exacerbating the concern of antimicrobial misuse. In this study, we observed that brand A, with a higher price, was the first-ranked efficacious drug (81% susceptibility of *S. aureus*) as compared to the other brands tested. Brand B, with a middle-value price, was the second-ranked brand in terms of efficacy (23% intermediate susceptibility of *S. aureus*), while brand C, with a price less than that of brands A and B, was found to be the least efficacious brand (1% intermediate susceptibility of *S. aureus*). Our study substantiated the fact that low-quality (substandard) medicines are prevalent in Sub-Saharan African countries [12,13], indicating that generic drugs commonly used by people in poor countries could promote the development of AMR by susceptible organisms.

The efficacy of different brands of penstrep tested in this study was directly related to the capacity of the drug (percent of inhibition for the test organism), i.e., the more the drug inhibits the test bacteria, the higher the efficacy of the drug [31]. Combining drugs could also have a synergist effect. This is true for the synergistic nature of a combined formulation of penicillin and streptomycin [32]. In this study, we found a better efficacy of a fixed combination form of penicillin and streptomycin—for instance, penstrep brand A (81%) over streptomycin (16%).

The variations in efficacy among different brands of antibiotics constitute a potential danger to health. Administering counterfeit drugs is known to result in therapeutic failure, toxicity, allergic reactions due to their content, development of AMR, prolonged illness, inflated costs of treatment, and even mortality, which all can directly or indirectly influence public health. Most developing and underdeveloped countries were reported to suffer from the direct and indirect effects of inferior-quality drugs to a high degree [33]. This is related to most societies found in such countries that can afford (high-use) low-cost drugs. As observed in this study, brands of drugs with low efficacy were imported or bought at a low price. Moreover, we noted that the livestock sector of the country buys imported drugs with the least price-based competitive bid system, perceiving all generic products to have equal clinical efficacy. Because of this, generic products with the lowest-priced antibiotics are commonly available in government veterinary clinics, but private clinicians prefer to use well-known brands. Hence, for the current study, it is very straightforward to predict the effect of the overuse of low-priced generic products or brand(s) of antibiotics. This could contribute to the emergence and re-emergence of increasing drug-resistant strains in resource-limited countries.

The observed differences in the efficacy of the three brands of penstrep in this study are believed to originate from the manufacturing process and internal composition (active pharmaceutical ingredients, API) or logistic systems. This constitutes a potential danger to animal health and welfare. This has clinical implications and impacts the treatment outcomes, as penstrep is commonly prescribed in all dairy farms in the study area for managing suspected systemic bacterial infections, wounds, and mastitis and as prophylaxis and metaphylaxis during FMD and LSD outbreaks. The predominant usage of penstrep in dairy farms is in agreement with another study previously conducted in Ethiopia [34].

Interestingly, the perception among veterinarians about the quality of the different brands of penstrep is perfectly in agreement with the antimicrobial susceptibility testing results. This study showed that the majority (70%) of veterinarians perceived that the quality of generic drugs is equivalent to that of brand drugs. However, they preferred prescribing drugs by brand names when there is a shortage of antibiotics in the government veterinary clinics, perceiving the availability of poor-quality generic products on the local market. Similarly, our study indicated that veterinarians perceived antibiotics produced in Western countries to have a better quality than those produced in Eastern countries. This study is in line with a report from Ghana on maternal healthcare products indicating the poor performance of “Indian-made” products but no quality problems for European-manufactured medicines [35]. In the present study, the prescriber’s perception towards the country of the antibiotic manufacturers shows that all veterinarians perceive penstrep produced in the UK as better in terms of quality compared with that produced in China. Similarly, all clinicians experienced differences in the clinical improvements of animals being treated with different brands of penstrep. Accordingly, all veterinarians preferred prescribing brand A because they frequently observed that it has better clinical effectiveness than the other two brands of penstrep. This finding is in line with the in vitro antibacterial efficacy study of the three brands against *S. aureus*, indicating variations in the amount of active ingredients in each brand, which contradicts the US Food and Drug Administration (FDA)’s suggestion that generic drug products must contain identical amounts of the same active drug ingredient as the brand name product [36].

The knowledge among clinicians about correct indications of major antibiotics is very crucial to improving animal husbandry practices, as the irrational prescription of antibiotics may perpetuate AMR [34,37]. The current study showed that veterinarians have good knowledge about major clinical indications of antibiotics. This is in line with the findings reported by a study conducted in India, where the judicious use of antibiotics was found to reduce antibiotic usage in farms and humans [38]. In contrast, there are reports of poor knowledge among veterinarians about antibiotic use in Southeast Asia (Timor-Leset) and among para-veterinarians in Fiji [39,40]. Our current study also revealed that veterinarians perceived that antibiotics are overused (widely prescribed) in veterinary clinics (87%) compared to dairy farms (53%). This is in line with our previous study conducted in Ethiopia, which showed that very limited antibiotic classes (tetracyclines, penicillins, aminoglycosides, and sulfa drugs) are widely prescribed and overused in veterinary clinics [19,27].

The current study has some limitations. First, we used an in vitro antimicrobial activity assay against only *S. aureus*, and the findings cannot be generalized for other pathogens. Second, due to the absence of standard clinical and epidemiological cut-off values for penstrep, the ZI and efficacy of the three brands were interpreted in reference to the cut-off values of penicillin and streptomycin. This suggests the need for further studies to establish and standardize clinical cut-off values for penstrep. Third, the concentration of brands of penstrep impregnated on Whatman paper may vary compared with the commercial penstrep disc (if available), as the sedimentation and dispersion of the suspended active ingredients (particles) of each brand may vary. This may have a great effect on efficacy evaluation in both in vitro testing and in vivo clinical efficacy using an animal model.

## 4. Materials and Methods

### 4.1. Study Area

The study was conducted from February to June 2018 in the towns of Bishoftu and Sebata of the central Oromia region, Ethiopia. Bishoftu is located in the East Shewa Zone of the Oromia regional state, and the area is located at a latitude of 40°E and a longitude of 8°45′ N 38°59′ E at an altitude of 1920 m above sea level in the central highlands of Ethiopia [41]. Sebata is located in the Oromia special zone surrounding Finfinnee (Addis Ababa) of the Oromia region, Ethiopia; it is approximately 25 km southwest of Addis Ababa at a latitude and longitude of 8°54′40″ N 38°37′17″ E and an altitude of 2356 m above sea level [42]. Farmers in the vicinity of Bishoftu and Sebata practice a mixed crop and livestock farming system, and smallholder dairy farming is common in the towns [43].

### 4.2. Study Design and Sampling

A cross-sectional study was conducted to determine the efficacy of brands of penstrep against *S. aureus.* To this effect, we conducted AST on *S. aureus* isolated from cow milk samples from smallholder dairy farms in Bishoftu and Sebata. Based on the location and willingness of farmers, a convenience sampling strategy was used to select 23 smallholder dairy farms (14 from Bishoftu and 9 from Sebata). About 20% of the lactating cows’ udders, including bulk milk from the farm, were considered for sampling. Accordingly, a total of 209 cows (186 randomly selected dairy cows and 23 bulk milk), 114 from Bishoftu and 95 from Sebata, were included in the study for the isolation of *S. aureus*.

The respondents were interviewed using a pretested and structured questionnaire (Appendix A) (translated to local languages; Afaan Oromo and Amharic), which was used to assess the KAP of veterinarians (*n* = 30) regarding the antibiotic quality and use of brands of antibiotics. The farm antibiotic utilization practices of dairy personnel (one person per dairy farm, *n* = 23) were also assessed. The respondents include veterinarians, animal health assistants, and farm managers/supervisors or farm owners. The questionnaire was designed based on previous studies [34] and categorized as constructs of (i) perception of veterinary antibiotics quality and brand prescribing (*n* = 6), (ii) knowledge and perception of antibiotics and their use (*n* = 9), and (iii) practices of antibiotic prescribing (*n* = 2). The questionnaire was developed to assess the antibiotic utilization practices of the farms: (i) antibiotics commonly used in the farm, (ii) the purpose of antibiotic use (prophylactic, treatment, or both), (iii) dairy animal diseases or symptoms commonly treated by antibiotics. In collaboration with the respondents, we also performed an inventory of available medicines or empty bottles and collected data from farm records (case book) (where available) during each farm visit for an interview. Both private and government employee clinicians were selected for the interview. A total of 23 farm managers/owners and 30 volunteer veterinarians, who practice drug prescribing in the government and/or private veterinary clinics or render ambulatory private clinical services in the study area, were included in the study.

### 4.3. Identifying Brands of Penstrep

Following the recommendation by Newton et al. (2009) for the assessment of the quality of medicines [44], three different brands of penstrep were purchased from a veterinary pharmacy located in Bishoftu. As per Newton et al.’s suggestion, we tried to determine the status of poor-quality medicine in the area. Accordingly, to know whether the medicines sold from a pharmacy are of poor quality or are legally imported or not, we randomly approached a pharmacy and asked questions to the pharmacy personnel: (1) “Are there medicines of poor quality in this geographical area?” (2) “Are there any outlets selling poor-quality medicines?” (3) “Is there any illegally imported antibiotics in this pharmacy?” Once we collected relevant information (“Yes”, “I don’t know”, and “No” responses to the three questions, respectively), the presence of brands of penstrep was asked about, finding the following three brands: Pen&strep (Norbrook), Penstrep (Chengdu Quiankun), and Pro&strep (Hebei Yuanzheng). We also asked, “What is the proportion of sales of each of these brands per day?” We collected feedback on the higher proportion of daily sales of Pen&strep (Norbrook); however, its availability is very limited due to its high price and is mostly prescribed by some private clinicians. Then, all brands were checked for proper storage and packaging and randomly sampled from different boxes. The three brands (Table 3) were legally registered and approved for use in veterinary medicine in Ethiopia. During the questionnaire survey, the brands of penstrep were selected by surveying their availability in the veterinary clinics and or pharmacies/drug stores in the study area, by assessing the most widely prescribed brands by veterinary clinicians, and by observing the most commonly found brands in the smallholder dairy farms. Details of the selected product characteristics such as the drug name, company name, country of production, batch number, legal permission, preparation, color inspection, manufacturing and expiry dates, and availability of label inserts were observed and recorded [45].

### 4.4. Isolation and Identification of S. aureus

The isolation and identification of *S. aureus* from milk samples were carried out following the Standard Operating Procedures recommended for microbiological techniques [46].

### 4.5. Efficacy Evaluation

#### 4.5.1. Preparation of Antibiotic Discs

We followed the method recommended by Vineetha et al. (2015) in preparing filter paper-based antibiotic discs. Discs 6 mm in diameter were punched from a sheet of Whatman number 1 filter paper by a perforator. The discs were wrapped in aluminum foil and sterilized in an oven at 160 °C for 15 min. After the discs were allowed to cool at room temperature, their sterility was confirmed by testing five randomly selected discs (free of antibiotics) by placing them on Mueller Hinton Agar (MHA) in a petri dish and incubating them at 37 °C for 24 h for any visible growth. The discs were used only when an absence of growth on the test discs was observed [47].

The standard commercial discs for both Penicillin-G (10 units) and Streptomycin (10 µg) were available commercially (Oxoid, UK). The known volume of different brands of penstrep was diluted at the time of disc preparation using sterile distilled water to obtain the working solution equivalent to the concentrations of the commercial standard discs. Tubes filled with sterile distilled water were used to carry out dilution until the concentration of the antibiotic solutions reached 10 µg (20 µL) [47].

The sterile discs were placed in petri dishes, and a fixed volume of 20 µL of the solution of each prepared brand of penstrep was loaded on each disc one by one using a pipette. The impregnated discs were arranged in separate plates and dried by placing them in an incubator at a temperature of 37 °C for 2 h [47].

#### 4.5.2. Antimicrobial Susceptibility Testing

The efficacy of the prepared antibiotic discs was tested against *S. aureus* using the recommended protocol: the Kirby–Bauer disc diffusion method. A day-old fresh subculture of presumptive *S. aureus* (*n* = 43) isolated from cow milk samples was used, and the concentration of the cultures was visually matched with 0.5 McFarland turbidity [5,48,49].

Three replicates of prepared antibiotic discs were aseptically placed on the MHA plates, along with the commercially available standard discs (Penicillin-G 10 unit and Streptomycin 10 µg), for a comparison of the efficacy of the prepared discs. The plates were then incubated at 37 °C overnight and measured for a diameter of zones of inhibition (ZI) using a digital caliper [5,28].

Penstrep has no specific standard ZI stated by CLSI or other standards. Hence, the interpretation of the results was performed as per the guidelines of the Clinical and Laboratory Standards Institute CLSI (2015) for penicillin and streptomycin. Accordingly, the isolates were classified as susceptible (S), intermediate (IR), and resistant (R) [50,51]. The mean ZI cut-off values for Penicillin (10 units) were ≥29 mm S and ≤28 mm R, whereas, for Streptomycin 10 µg, we used the value set for Gentamycin ≥15 mm S, 13–14 mm IR, and ≤12 mm R [50]. However, IR was recorded as resistant. As an in vitro efficacy evaluation for different brands of penstrep against *S. aureus* has not been conducted so far, the specific zone of inhibition (ZI) for different brands of penstrep was compared with both the penicillin and streptomycin standards, as both drugs were combined in equal concentrations, but not for brand A. Both the microbiological analysis and efficacy evaluation were performed at Addis Ababa University College of Veterinary Medicine and Agriculture (AAU-CVMA), Bishoftu.

### 4.6. Statistical Analysis

Data were entered into a Microsoft Office 365 Excel spreadsheet and analyzed using R software Version 3.4 (R Foundation for Statistical 192 Computing, Vienna, Austria). The study outcomes were the efficacy of the tested brands of antibiotics and the KAP of the respondents. The efficacy of each brand was determined by the level of susceptibility of the tested isolates (either resistant or susceptible). Categorical variables were expressed by proportions, and their significance was assessed, when appropriate, using a Chi-square (χ^2^) test or Fisher’s exact test. This is used to test for potential associations between categorical variables, and a *p*-value of ≤0.05 was considered statistically significant. Continuous variables were expressed by means and standard deviations (SD) and assessed for statistical significance using the Kruskal–Wallis chi-square test. Descriptive statistics such as frequency (percentage) of resistance were computed to summarize a dataset for the efficacy evaluation of different brands of penstrep. The results of the ZI recorded by the tested antibiotics and interpreted as an IR were expressed by the value of resistance (R). The measured value for the ZI of the three different brands was compared with the corresponding standards for penicillin and streptomycin.

Descriptive statistics were also used to compute the results of farm antibiotic utilization practices. The results of the KAP of veterinarians regarding veterinary antibiotics quality and brand prescribing questions were expressed by the values of either 0 (not correct) or 1 (correct), and the degree of their agreement was measured on a five-point Likert scale from “Strongly agree/agree” to “Strongly disagree /disagree.” A “Neutral” answer was not recorded in this study. The percentage of correct answers for each knowledge question and the statements “Strongly agree/Agree” by the respondents were calculated.

## 5. Conclusions

The study revealed variability in the efficacy of different brands of penstrep available in Ethiopia. Comparatively, brand A was found to be a more effective drug than other brands. Intuitively, veterinarians’ perception of prescribing brand antibiotics also agrees with the in vitro antibacterial efficacy evaluation findings. The availability of ineffective brands constitutes a potential danger to both human and animal health and welfare due to therapeutic failure, AMR, prolonged illness, the high cost of treatment, and even death. Further large-scale, robust studies that focus on establishing a resistance cut-off value for penstrep, the evaluation of efficacy regarding other bacteria, an in vivo study in animal models, and stringent regulation and quality assays for active pharmaceutical ingredients of different brands of penstrep are critically required.

## Figures and Tables

**Figure 1 antibiotics-12-00570-f001:**
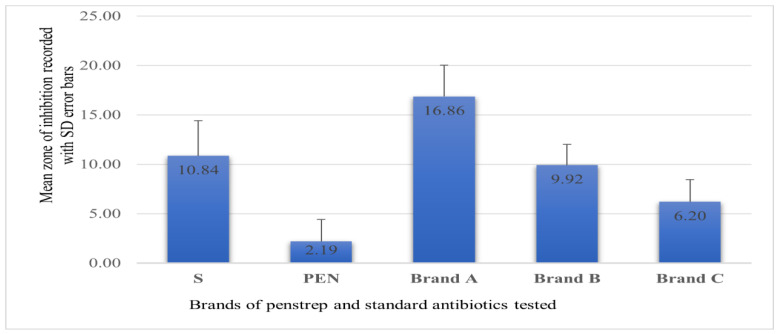
Zone of inhibition of the three brands of penstrep and standard discs against *S. aureus* (*n* = 43) isolated from dairy cow milk in Bishoftu and Sebata. (S, Streptomycin; PEN, Penicillin-G; SD, standard deviation; ZI, zone of inhibition.)

**Table 1 antibiotics-12-00570-t001:** Antibacterial efficacy of three brands of penstrep against *S. aureus* (*n* = 43) isolated from dairy cow milk in Bishoftu and Sebata, Ethiopia.

Drugs Tested	Susceptibility of the Isolates		
	Resistant *n* (%)	IR *n* (%)	Susceptible *n* (%)
Penicillin (P)	43 (100.0)	0 (0.0)	0 (0.0)
Streptomycin (S)	27 (63)	9 (21)	7 (16)
Brand A compared to (P) *	43 (100)	0 (0.0)	0 (0.0)
Brand A compared to (S) ^#^	4 (9)	4 (9)	35 (81)
Brand B compared to (P) *	43 (100)	0 (0.0)	0 (0.0)
Brand B compared to (S) ^#^	33 (77)	10 (23)	0 (0.0)
Brand C compared to (P) *	43 (100)	0 (0.0)	0 (0.0)
Brand C compared to (S) ^#^	42 (98)	1 (2)	0 (0.0)

*n*, the number of isolates; %, percent; P, penicillin (10 µg); S, streptomycin (10 µg); * indicates the susceptibility test result for the three brands as compared to the standard set for penicillin; # indicates the susceptibility test result for the three brands as compared to the standard set for streptomycin. IR, Intermediate resistance was considered as resistance.

**Table 2 antibiotics-12-00570-t002:** Veterinarians’ perception, knowledge, and practices of antibiotic quality, use, and brand prescribing in Bishoftu and Sebata (2018).

Question	Participants’ Agreement/Total Participants (%)
*Perception of veterinary antibiotics quality and brand prescribing*	
Agree that some antibiotics on the market are of poor quality	30/30 (100)
Penstrep imported from a Western country (e.g., the UK) is perceived as of better quality than those from Eastern countries (e.g., China)	30/30 (100)
Generic antibiotics are perceived as of equivalent quality compared to branded antibiotics	21/30 (70)
Generic antibiotics are perceived as substandard drugs	9/30 (30)
Prescribe antibiotics by international nonproprietary name	14/30 (80)
Prescribe antibiotics by brand name	20/30 (67)
Prescribe brands of penstrep	26/30 (87)
Variation in clinical improvements among brands of penstrep	30/30 (100)
Which brands of penstrep showed better clinical improvements?	
Pen&Strep (Norbrook)	30/30 (100)
Which brands of penstrep are mostly prescribed?	
Pen&Strep (Norbrook)	30/30 (100) ^a^
Penstrep (Chengdu Quiankun)	16/30 (53) ^b^
Pro&Strep (Hebei Yuanzheng)	8/30 (27) ^c^
*Knowledge of antibiotics and their use*	
Penstrep indicated for both Gram-positive and Gram-negative bacterial infections	30/30 (100)
Oxytetracycline indicated for gastro-intestinal and respiratory bacterial infections	30/30 (100)
Sulfa drugs indicated for diarrheic cases	30/30 (100)
Antibiotics indicated for prophylactic use for severe viral cases	30/30 (100)
*Antibiotic prescribing practices*	
Perception of antibiotic overuse in the dairy farms	16/30 (53)
Perception of antibiotic overuse in the veterinary clinics	26/30 (87)

^b,c^ prescribed only when the drug ^a^ was not available in the clinic.

**Table 3 antibiotics-12-00570-t003:** Characteristics by strength, country of origin, and average price of brands of penstrep used for efficacy evaluation against *S. aureus*.

Brand Name	Concentration (mg/mL)	Country of Origin	Average Price per Bottle of 100 mL in ETB (USD)
Brand A	S (250 mg/mL) P (200 mg/mL)	UK	500 (9.615)
Brand B	S (200 mg/mL)	China	380 (7.307)
	P (200 mg/mL)		
Brand C	S (200 mg/mL)	China	350 (6.730)
	P (200 mg/mL)		

ETB, Ethiopian birr; S, Streptomycin; P, Penicillin; $ the current average selling price from retailers was used for comparison (USD 1 = ETB 52.0 in June 2022).

## Data Availability

The datasets used and/or analyzed during the current study are available in the manuscript. All data supporting the reported results can be found in publicly archived datasets generated during the study and uploaded to a preprint as supplementary files [52].

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
