# Peer review of "Efficacy of Penicillin–Streptomycin Brands against Staphylococcus aureus: Concordance between Veterinary Clinicians’ Perception and the Realities"

_antibiotics, 2023, doi:10.3390/antibiotics12030570_

Round 1
Reviewer 1 Report
The manuscript entitled “Efficacy of penicillin-streptomycin brands against Staphylococcus aureus: Concordance between clinicians’ perception and the realities” is well written and properly explained. Authors of this article have raised important yet neglected sector in public health about rational use of antibiotics in veterinary practice especially in developing and underdeveloped countries. Antibiotics are our important line of defense against bacterial infections and their irrational use in veterinary and public health has poorly affected the efficacy of commercially available drugs. It is a major issue in developing countries and is alarming for whole world due to development of resistant strains of bacteria. Availability, efficacy, trust and knowledge are the important factors, which guide the clinician to use the antimicrobial drugs. These issues have been comprehensively discussed in this manuscript. Common brands of antibiotics available in Ethiopia are tested in this study, which is important for local veterinarians and government officials to guide them to make effective plans for the rational use of antibiotics and decision-making process while selecting an antimicrobial drug for common infections and diseases. This study will also help the local livestock community by improving their trusts on local veterinarian as the invitro finding are in accordance with the perceptions of veterinarians. Furthermore, this study has indicated that local veterinary community is well aware of rational use of antibiotics, however their availability and price is the main problem faced by local livestock community. Shortcomings and limitations of this study are well addressed and explained. Future studies are also proposed and discussed in details which make this manuscript complete. Regards,
Author Response
Dear Reviewer,
Thank you very much for dedicating your precious time and providing constructive comments and suggestions for further improving our manuscript no. antibiotics-2259276. We also appreciated and thankful for the good explanation about issues of antibiotic misuse in livestock in the least developed countries. We provided step by step response to each of your query as indicated below:
Point 1. English language and style are fine/minor spell check required
Response 1. We acknowledge the observation made by the reviewer and we have addressed the issues by making thorough editions for the English language correcting spelling and grammatical errors and further improved the manuscript by editing for contents. The manuscript has also been reviewed and edited by co-authors residing and working in the USA. Kindly see the attached mark-up file for the edition and revision made.
Point 2. The manuscript entitled “Efficacy of penicillin-streptomycin brands against Staphylococcus aureus: Concordance between clinicians’ perception and the realities” is well written and properly explained. Authors of this article have raised important yet neglected sector in public health about rational use of antibiotics in veterinary practice especially in developing and underdeveloped countries. Antibiotics are our important line of defense against bacterial infections and their irrational use in veterinary and public health has poorly affected the efficacy of commercially available drugs. It is a major issue in developing countries and is alarming for whole world due to development of resistant strains of bacteria. Availability, efficacy, trust and knowledge are the important factors, which guide the clinician to use the antimicrobial drugs. These issues have been comprehensively discussed in this manuscript. Common brands of antibiotics available in Ethiopia are tested in this study, which is important for local veterinarians and government officials to guide them to make effective plans for the rational use of antibiotics and decision-making process while selecting an antimicrobial drug for common infections and diseases. This study will also help the local livestock community by improving their trusts on local veterinarian as the invitro finding are in accordance with the perceptions of veterinarians. Furthermore, this study has indicated that local veterinary community is well aware of rational use of antibiotics, however their availability and price is the main problem faced by local livestock community. Shortcomings and limitations of this study are well addressed and explained. Future studies are also proposed and discussed in details which make this manuscript complete.
Response 2: Thank you for your critical review.
Reviewer 2 Report
This paper is of significant interest in that it provides documented evidence of variations in commercial veterinary antibiotic quality that have long been suspected but not examined to a great extent by scientific evaluation and published, particularly for countries in E Africa. It draws attention to the many sources of variation in the quality of different antibiotic preparations and hence their relative efficacy (in this case in vitro). The paper is generally well written, although some English language editing is necessary, and it is apparent that the study was diligently conducted as methodologies were carefully referenced.
My main question concerns Ethical review. Could the authors please provide a statement on the ethical requirements in Ethiopia for such a study and how this was dealt with.
One specific point: In the Materials and Methsds, the reference of Newton et al 2009 is cited for the assessment of quality of medicines (ll308-309). A bit more xplanation of this in the present text would be useful here.
Author Response
Dear Reviewer,
Thank you very much for dedicating your precious time and providing constructive comments and suggestions for further improving our manuscript. (no. antibiotics-2259276). We provided step-by-step responses to each of your queries as indicated below.
Point 1. This paper is of significant interest in that it provides documented evidence of variations in commercial veterinary antibiotic quality that have long been suspected but not examined to a great extent by scientific evaluation and published, particularly for countries in E Africa. It draws attention to the many sources of variation in the quality of different antibiotic preparations and hence their relative efficacy (in this case in vitro). The paper is generally well written, although some English language editing is necessary, and it is apparent that the study was diligently conducted as methodologies were carefully referenced.
Response 1: Thank you for your critical review.
Point 2. Quality of English language: Moderate English changes required.
Response 2. We acknowledge the observation made by the reviewer and we have addressed the issues by making thorough edition for the English language correcting spelling and grammatical errors and further improved the manuscript by editing for contents. The manuscript has also been reviewed and edited by co-authors residing and working in the USA. Kindly see the attached mark-up file for the edition and revisions made.
Point 3. Are the results clearly presented? (Answer: can be improved)
Response 3. We appreciate the remarks made by the reviewer. The result section and the other parts of the manuscript are further revised. Kindly see the changes made in the mark-up file.
Point 4. My main question concerns Ethical review. Could the authors please provide a statement on the ethical requirements in Ethiopia for such a study and how this was dealt with.
Response 4. We appreciate the concern raised by the reviewer. In Ethiopia, Institutional Review Board (IRB) approval is required for any study involving animals and humans. Our study involves both animals and human subjects. Data were generated by interviewing farm owners and veterinarians and collecting milk samples from dairy cows to isolate S. aureus for testing and evaluating the efficacy of the three brands of the Penstrep. To that end, our proposal was submitted to AAU_ IRB, reviewed, and approved as indicated in the manuscript (Page 112 Line 424-431).
To provide a context about our ethical approach, during the enrollment of study participants, we explained to them the purpose of the research and let them participate in the study voluntarily after obtaining verbal consent. For privacy issues, we informed them that their names or any other identifiers will not be shared with anyone and will not be mentioned in the publication of the results as individual participants. However, data generated from them will be included in the publication. In Ethiopia, in most cases, farmers do not want to sign a consent letter, but they want to keep their consent verbally. For studies involving animals, milk sample collection was a non-invasive type and there is no animal ethics concern. Regarding the efficacy results of the brand Penstrep, we opt not to mention the name of the pharmaceutical companies in the manuscript; however, we mentioned the name of the brands as Brand A, Brand B, and Brand C for comparison purposes. This issue was a discussion point even among the authors. Finally, we all agreed not to mention explicitly the details of the companies and recommend the government to further conduct post-marketing surveillance on all brands of antibiotics.
Point 4. One specific point: In the Materials and Methods, the reference of Newton et al 2009 is cited for the assessment of quality of medicines (ll308-309). A bit more xplanation of this in the present text would be useful here.
Response 4. We agree with the remark made by the reviewer and the issue has been addressed in the revised version of the manuscript (Pages 9-10)
Reviewer 3 Report
This paper is interesting and the work is carefully done. It is particularly interesting in this period the study and application of antimicrobial stewardship, especially in an African country that normally are hardly involved in these studies. As expected and unfortunately the better formulation of combination pen+strep are that produce in Europe, branded and with a higher cost.
I completely agree with the conclusions of this article. It is a battle to be fought in every part of the world and it is particularly important that paper like this arrrived from this part of the world.
Some suggestions:
There are a mistake at line 61: the study described is NOT published in Lancet, but on Scientific America.
Line 345: what is the difference between 20 ul and 0.02 ml?
Please write correctly the Reference n. 44
Author Response
Dear Reviewer,
Thank you very much for dedicating your precious time and providing constructive comments and suggestions for further improving our manuscript (no. antibiotics-2259276). We provided step-by-step responses to your queries as indicated below.
Point 1: This paper is interesting and the work is carefully done. It is particularly interesting in this period the study and application of antimicrobial stewardship, especially in an African country that normally are hardly involved in these studies. As expected and unfortunately the better formulation of combination pen+strep are that produce in Europe, branded and with a higher cost.
I completely agree with the conclusions of this article. It is a battle to be fought in every part of the world and it is particularly important that paper like this arrived from this part of the world.
Response 1: Thank you very much for the critical evaluation.
Point 2: English language and style are fine/minor spell check required.
Response 2: We acknowledge the observation made by the reviewer and we have addressed the issues by making thorough editions for the English language: correcting spelling and grammatical errors and further improved the manuscript by editing for contents. The manuscript has also been reviewed and edited by co-authors residing and working in the USA. Kindly see the attached mark-up file for the edition and revision made.
Point 3. Are all the cited references relevant to the research? Answer: Can be improved.
Response 3. We appreciate the remarks made by the reviewer. We believe we cited all relevant literature to support our findings.
Some suggestions:
Point 4: There are a mistake at line 61: the study described is NOT published in Lancet, but on Scientific America.
Response 4. We appreciate the remarks made by the reviewer and we have made updates. The appropriate works published in The Lancet were cited.
Point 5: Line 345: what is the difference between 20 ul and 0.02 ml?
Response 5. We noted the remarks of the reviewer. There is no difference between 20 ul and 0.02 ml. We have addressed the issue and used 20 µl (page 10)
Point 6: Please write correctly the Reference n. 44
Response 6. We appreciate the remarks made by the reviewer and we have addressed the issue. (Page 15)